# Using Serological Markers for the Surveillance of *Plasmodium vivax* Malaria: A Scoping Review

**DOI:** 10.3390/pathogens12060791

**Published:** 2023-05-31

**Authors:** Lejla Kartal, Ivo Mueller, Rhea J. Longley

**Affiliations:** 1School of Population and Global Health, The University of Melbourne, Parkville 3010, Australia; lkartal@student.unimelb.edu.au; 2Population Health and Immunity, The Walter and Eliza Hall Institute of Medical Research, Parkville 3052, Australia; mueller@wehi.edu.au; 3Department of Medical Biology, The University of Melbourne, Parkville 3010, Australia

**Keywords:** *Plasmodium vivax*, malaria, serological markers, antibodies, surveillance, serology, humoral immunity, serosurveillance

## Abstract

The utilisation of serological surveillance methods for malaria has the potential to identify individuals exposed to *Plasmodium vivax*, including asymptomatic carriers. However, the application of serosurveillance varies globally, including variations in methodology and transmission context. No systematic review exists describing the advantages and disadvantages of utilising serosurveillance in various settings. Collation and comparison of these results is a necessary first step to standardise and validate the use of serology for the surveillance of *P. vivax* in specific transmission contexts. A scoping review was performed of *P. vivax* serosurveillance applications globally. Ninety-four studies were found that met predefined inclusion and exclusion criteria. These studies were examined to determine the advantages and disadvantages of serosurveillance experienced in each study. If studies reported seroprevalence results, this information was also captured. Measurement of antibodies serves as a proxy by which individuals exposed to *P. vivax* may be indirectly identified, including those with asymptomatic infections, which may be missed by other technologies. Other thematic advantages identified included the ease and simplicity of serological assays compared to both microscopy and molecular diagnostics. Seroprevalence rates varied widely from 0–93%. Methodologies must be validated across various transmission contexts to ensure the applicability and comparability of results. Other thematic disadvantages identified included challenges with species cross-reactivity and determining changes in transmission patterns in both the short- and long-term. Serosurveillance requires further refinement to be fully realised as an actionable tool. Some work has begun in this area, but more is required.

## 1. Introduction

Malaria is one of the top ten causes of death in low-income countries [1]. It is caused by the mosquito-borne *Plasmodium* parasite, of which there are at least five species that infect humans: *P. falciparum*, *P. vivax*, *P. malariae*, *P. ovale*, and *P. knowlesi* [2,3]. *P. falciparum* and *P. vivax* account for most of the disease burden globally, although their respective distributions differ [3]. *P. falciparum* is largely prevalent in sub-Saharan Africa, whereas *P. vivax* is more prevalent in the Americas, Asia and the Pacific [3,4]. The World Health Organization has set the explicit goal of achieving malaria elimination in at least 35 countries by 2030, and malaria caused by *P. vivax* stands to be a formidable obstacle to achieving this goal [5]. *Plasmodium* spp. is typically detectable by blood-based diagnostic methods, including gold-standard microscopy; however, due to its low parasitaemia, *P. vivax* is significantly more difficult to diagnose with these methods [2,4,6]. For malaria elimination to be feasible, the reservoir of asymptomatic cases must be identified and treated, which is not possible with current diagnostics [7,8,9]. Whilst ultra-sensitive PCR-based methods can identify blood-stage infections with low peripheral parasitaemia, a large biomass of *P. vivax* parasites is now known to be present in organs such as the spleen and bone marrow [10,11]. In addition, *P. vivax* has an arrested liver stage, hypnozoites, that lead to relapsing malaria infections (Figure 1). Individuals with hypnozoites show no clinical signs of infection, and hypnozoites are not detectable with current diagnostics. These parasites can remain within the liver for many months-years and contribute to as much as 80% of all *P. vivax* blood-stage infections (versus new mosquito bite-induced infections) [12].

One possible strategy to overcome the limitations of blood-stage antigen detection-based diagnostics is the use of serology to identify individuals who have generated antibodies to *P. vivax*, as this can serve as an indirect marker of exposure to *P. vivax*. This method consists of screening individuals for antibodies to chosen *P. vivax* protein antigens as opposed to screening for the presence of the parasite itself, thereby inferring from seropositivity that an individual has previously been exposed to *P. vivax* [3]. Surveillance is critical for malaria elimination, as it can inform on local transmission and highlight areas in need of further targeting [14]. However, if surveillance systems are unable to identify asymptomatic individuals (those with low-density infections, cryptic infections, or hypnozoites), their capacity to inform on transmission patterns and events may be undermined [15]. Serology offers an auspicious tool to supplement standard surveillance methods by facilitating the identification of asymptomatic individuals who may be missed by other diagnostic methods. Serosurveillance has been used for many other pathogens globally, including Ebola virus [16], *Chlamydia trachomatis* [16,17], human immunodeficiency virus (HIV) [16], hepatitis B virus [18], lyssavirus (rabies) [19], dengue virus [20,21], and SARS-CoV-2 (COVID-19) [22]. Multiple methods exist for the measurement of antibodies, many of which are currently in use for the serological surveillance of *P. vivax*; however, the implementation of these methods varies widely, particularly as *P. vivax* expresses approximately 5000 proteins [15,23,24]. These proteins are expressed across various stages of the parasite’s lifecycle, including the dormant hypnozoite stage as well as the blood stage (see Figure 1) [25,26]. The use of serology enables the identification of exposed individuals without relying on typical blood-stage diagnostics. 

This scoping review seeks to probe the currently published literature in the field of serosurveillance for *P. vivax*, explore how it has been applied, in what contexts, and identify the challenges and accomplishments that have been encountered during implementation.

## 2. Materials and Methods

To ensure all published instances of serological testing being utilised for the surveillance of *P. vivax* were included, the scoping review methodology was chosen in accordance with the PRISMA Extension for Scoping Reviews (PRISMA-ScR): Checklist and Explanation [27]. This presents a more rigorous approach to *P. vivax* serosurveillance review than has been previously performed. The review protocol is presented below in full.

### 2.1. Search Strategy

The following search strategy was utilised to identify relevant peer-reviewed studies. One electronic database (PubMed) was searched. The literature search terms were vivax AND serology OR serosurveillance OR sero-surveillance OR sero surveillance OR surveillance OR antibod*. The truncated term antibod* was utilised to capture both “antibody” and “antibodies” as search terms. No date restriction was utilised. Search results were imported into Covidence [28] for screening. The fully completed PRISMA-ScR checklist is included in Appendix A.

### 2.2. Eligibility Criteria

All peer-reviewed studies identified in the search were screened for eligibility. Eligible studies were published in English, performed with humans, included primary research, had the full text available, and explored the use of serology for *P. vivax*. Studies that included multiple *Plasmodium* species were included if *P. vivax* was distinct from other species.

### 2.3. Study Selection

Study screening and selection were performed using Covidence [28]. Duplicate studies were automatically screened for and removed. LK screened titles and abstracts of all unique identified studies and then screened the full text of all potential studies identified after abstract screening. The list of included studies was independently reviewed by RL to confirm all studies adhered to the eligibility criteria. Reasons for the exclusion of studies were recorded and reported in Covidence [28]. See Appendix A for the full list of included studies.

### 2.4. Data Extraction

Covidence [28] was used for data extraction. A customised data extraction template was generated to capture the following information: title, year of publication, author/s, the country in which the study was conducted, study design, participant demographic information, the total number of participants, serological markers used, type of antibody measured, serological test method/s used, whether new or historical samples were used (and from which year), serological results, advantages and disadvantages of serology that were highlighted by the study, and any additional commentary deemed relevant. Data were then analysed in accordance with relevant themes identified from the results. 

## 3. Results

### 3.1. Literature Search

The database searches were performed on 29 March 2022 and identified 1259 records, of which 583 were unique studies. The titles and abstracts of these studies were screened, leaving 238 potential studies. The full texts were assessed for eligibility; 94 studies satisfied the inclusion criteria (Figure 2).

### 3.2. Study Characteristics 

The included studies reported data from 34 different countries, including Brazil (*n* = 19), Cambodia (*n* = 6), China (*n* = 2), Ethiopia (*n* = 7), Indonesia (*n* = 2), Iran (*n* = 2), Republic of Korea (*n* = 17), Myanmar (*n* = 7), Senegal (*n* = 2), Thailand (*n* = 7), Vanuatu (*n* = 2), and Vietnam (*n* = 3) (Appendix A). One study was conducted in non-malaria endemic areas through the recruitment of recently returned travelers [29]. Note that some studies reported data from multiple countries. Sample sizes of the included studies ranged from 25 to 28,681 [30] participants. Study populations varied from people with clinical infections to entire populations of specific regions regardless of symptoms. Nine different antibody detection methodologies were used. Fifty-three studies (56.4%) used an enzyme-linked immunosorbent assay (ELISA), 12 (12.8%) used a multiplex bead assay, and 9 (9.6%) used an indirect fluorescent antibody test (IFAT) or immunofluorescent assay (IFA). The number of *P. vivax* antigens used ranged from 1 to over 1000 [8,31]. The most used antigens were the blood-stage expressed *P. vivax (Pv)* MSP1-19 (in >69% of studies) and *Pv* AMA1 (>31%), and the sporozoite-stage expressed *Pv* CSP (>29%). Over 82% of studies analysed IgG responses, whereas less than 10% of studies analysed IgM responses. 87.5% of studies that analysed IgM responses also analysed IgG responses (Appendix A). Nearly half (48.9%) of studies utilised the same method to set seropositivity cut-offs: mean +3×(standard deviation) of the negative controls. A further 7.4% used a very similar cut-off: mean +2×(standard deviation) of the negative controls. 13.8% of studies used various other cut-offs, including using 3×(mean) of the negative controls and categorising seropositivity using ordinal categories. 

### 3.3. Seroprevalence

The seroprevalence results are summarised in Table 1. Seroprevalence results were available for 73 studies from the years 2000–2022. The remaining studies either explored the performance of their chosen methodology [31,32,33,34,35,36,37,38,39,40,41,42] or measured the magnitude of antibody responses without reporting seroprevalence estimates [6,8,23,43,44,45]. One study utilised seropositivity as an exposure variable but did not report specific seroprevalence results [46]. Another study utilised serology to determine geographical ‘pockets’ of seropositivity but did not report seroprevalence estimates for these pockets [47]. Seroprevalence ranged from 0% [48,49,50] to 93.4% [51] across 31 countries. For studies reporting results utilising multiple antigens, both the minimum and maximum were considered for the overall seroprevalence range reported here. By decade, seropositivity ranged from 0% [48,49,52] to approximately 70% [53,54] in the 2000s, 0.45% [55] to approximately 80% [56,57] in the 2010s, and from 0% [30,58] to 93.4% [51] from 2020. By method, seropositivity ranged from 0% [48,49,52] to 93.4% [51] using ELISA, from <1% [30,58] to approximately 50% [59,60] using multiplex bead assays, and from 0% [52,61] to approximately 45% [62] using IFAT. Seropositivity also ranged by region. In Africa (Benin, Cameroon, Djibouti, Ethiopia, Guinea, Mali Republic of the Congo, Senegal, and Somalia), results ranged from 0% [49,58] to 57.4% [49]. In the Americas, it ranged from 0% [52] to 93.4% [51]. Results in Asia ranged from 0% [48] to approximately 80% [56,57]. Characteristics of the studies are presented in Appendix A.

### 3.4. Emerging Themes on the Advantages and Disadvantages of Serology Implementation

The articles included in this scoping review presented recurring themes. Broadly, these themes can be separated into advantages and disadvantages regarding the implementation of serology. 

### 3.5. Advantages

Highlighted advantages of utilising serology included ease of identifying exposed individuals [61,63,64,65,66], the ability to detect longitudinal and recent exposures [29,38,39,63], and the resources saved compared to the gold-standard light microscopy [29,32,33,67,68]. Serological methods were able to identify individuals who had been exposed to *P. vivax* with high sensitivity, particularly in areas of low and/or declining malaria transmission [61,63,64,65]. This was deemed to be due to most individuals with *Plasmodium* parasites forming anti-*Plasmodium* antibodies, even with low parasitaemia [66]. Anti-*Plasmodium* antibodies were largely species-specific between *P. vivax* and *P. falciparum* and therefore allowed for some distinction of *Plasmodium* species in individuals with multi-species infections [6,30,37,54,59,69,70,71]. However, cross-reactivity was noted in other studies, as detailed below in “*disadvantages*”. Additionally, the use of serology allowed asymptomatic individuals to be identified with more precision than other blood-based diagnostics [8,63,68,72,73,74,75,76]. This, in turn, provided more accurate prevalence estimates [29,77]. 

A significant correlation between the seroprevalence estimates and reported annual incidence of *P. vivax* malaria from the national malaria control programs was observed, indicating that antibody detection provides a longitudinal, rather than cross-sectional, assessment of malaria [48,58]. As a result, serosurveillance was useful for recording both longitudinal and recent exposures [29,38,39,63], pinpointing transmission foci [29,38,39,63], and monitoring transmission intensity [74]. Furthermore, serological results highlight groups at higher risk of malaria, as well as individual risk factors associated with malarial seropositivity [74,76,78]. Specifically, this was done through the calculation of age-specific seroprevalence rates and seroconversion rates [8,63]. Elimination of malaria was also evidenced using these results [8,79,80,81,82]. 

Serological methods were found to be inexpensive, faster, less labour-intensive, and simpler than the gold-standard microscopy method [29,32,33,67,68]. Serological methods can be further simplified for use in point-of-care/contact settings using rapid serological tests or dot-ELISAs, both of which can theoretically be performed and interpreted in the field without specialist resources [3,33,66]. Additionally, blood samples can be taken onto filter paper, dried, and stored until they can be delivered to testing laboratories if the resources are not available locally [83]. This method was highlighted as a means to assess retrospective transmission based on historical samples [84,85]. It was also noted that only one blood spot sample per individual is required to estimate transmission history for multiple species, thereby reducing the sampling burden when multiple surveillance systems exist [39,67,86]. 

### 3.6. Disadvantages

Highlighted disadvantages included cross-species reactivity [6,41,65,70,78,80,87,88], difficulty selecting appropriate antigens and antibodies [60,73,78,80,81,85,89,90], a lack of method standardisation [23,69,75], and varying applicability to differing epidemiological contexts [23]. Although anti-*Plasmodium* antibodies were largely species-specific between *P. vivax* and *P. falciparum*, some inter-species cross-reactivity was observed, particularly between *P. vivax* and other non-*falciparum* species (*P. knowlesi*, *P. ovale*, and *P. malariae*) [6,41,65,70,78,80,87,88]. This presented issues with both result interpretation and antigen selection [6,41,70,78,80,87,88,91]. It was found that cross-reactivity may occur due to sequence homology among proteins of different *Plasmodium* species; *P. vivax* is more closely related to *P. ovale*, *P. malariae*, and *P. knowlesi* than *P. falciparum*, which may lead to more cross-reactivity with these three species than *P. falciparum* [41]. It was commonly found that the inclusion of multiple *Plasmodium* antigens improved detection capacity [60,73,78,80,81,85,89,90]; however, mixed conclusions were drawn on the three most commonly used antigens. *Pv*MSP1-19 was deemed both useful [23,34,55,69,77,92,93] and uninformative [94] as an antigen to indicate exposure to *P. vivax*. *Pv*AMA1 was found to be both suitable [47,55] and ambiguous [94]. Kattenberg et al. [63] found *Pv*AMA1 to only be useful for long-term changes in exposure, as opposed to identifying recent exposure. *Pv*CSP was deemed both apt [35,57,66,92] and imprecise [52,74,80,93,95] for the identification of *P. vivax* exposure. Other issues compounded this difficulty in antigen selection, including the existence of polymorphisms for some proteins [56,88]. Furthermore, genetic diversity among antigens may inhibit or alter host antibody responses [15,41,56], which introduces complexity in the widespread use of identified antigens, as genetic diversity may be geographically structured [96]. Variability between individuals’ antibody responses and seroconversion times also added complexity, as the timing, magnitude, and waning of these responses all differed, meaning serological testing had to be carefully timed [23,33,51,70,81,90]. 

Numerous serological test methods were used to assess exposure to *P. vivax*; however, they were largely unvalidated and not standardised across geographic and transmission settings [23,69,76]. According to the authors, methodology, including protocols, antigen selection, recombinant antigen production, antibody detection, and cut-off point determination, all require standardisation to improve the reliability and comparability of results and studies [23,46,63,69]. 

Finally, serological testing was not found to be particularly useful for diagnosis but was used to detect previous exposure to *P. vivax* [64]. Serosurveillance assessed historical and recent transmission as opposed to current prevalence [67]; consequently, it was found to be more pertinent in low-transmission settings than in high-transmission [23]. However, it was noted that if antigens with a short-lived response are specifically chosen, serology could be used to infer recent rather than historical exposure [41]. This prompted the possibility of a novel public health approach through identifying individuals with recent exposure and providing relevant treatment [41].

## 4. Discussion

Areas of the world that are working towards the elimination of *P. vivax* must contend with the increasing burden of individuals with asymptomatic blood-stage infections and asymptomatic hypnozoite carriers. Identifying these individuals is complex as they are unlikely to seek treatment and typically have low parasitaemia (or no peripheral parasitaemia in the case of hypnozoite carriers), making standard diagnostic approaches impractical [29,97]. Striving for malaria elimination without the ability to readily detect these individuals is challenging, and therefore blanket approaches such as mass drug administration programs, which can be expensive, complex, and require extensive resources, are often used [98]. Serological methods can fill this gap by identifying individuals who have been exposed to *P. vivax*, thereby narrowing the pool of individuals receiving treatment to those most likely to need it [41,99]. However, there is considerable heterogeneity in serosurveillance methodology, and various factors need to be considered to ensure the most suitable approach is chosen for any given transmission setting. As the methodology for *P. vivax* serosurveillance is not standardised, it is important to explore and evaluate the currently used methodologies to identify areas of success, as well as areas for improvement, across various transmission contexts. No prior published scoping or systematic reviews on the merits and drawbacks of current approaches were identified. Of note was a recent review by Tayipto et al. [3], which illustrated challenges for malaria surveillance and considerations for the use of serosurveillance in the context of *P. vivax* elimination but did not explore the current state of serosurveillance more broadly. In this scoping review, 94 studies were identified in which authors described the seroprevalence of *P. vivax* and/or the advantages and disadvantages of their chosen serological method. Seroprevalence was found to vary widely, from 0% [48,49,50] to over 90% [51]. When split by decade, there appears to be no significant difference in seropositivity rates. However, it is important to note that this observation may be confounded by various factors that have not been explored here; further analysis would be required for confirmation. Overall, this review revealed that serological approaches varied widely across studies, both in terms of methodology and application. Several advantages and disadvantages of utilising serology were identified. Advantages of serology implementation included the ability to: identify exposed individuals; record both longitudinal and recent exposures; monitor the elimination of malaria; increase efficiency when compared with traditional light microscopy; and utilise dried blood spots. Disadvantages of serological approaches included: difficulties with antigen selection; variable applicability across epidemiological contexts; lack of method standardisation; and cross-reactivity between *Plasmodium* species. 

Antibodies are incredibly useful indicators for malaria transmission as they remain detectable after the infection has passed, therefore allowing a wider time period in which a blood sample can be taken and still indicate that an individual has been exposed to *P. vivax* [100]. Additionally, some antibody responses increase with age, thereby becoming a marker of lifetime exposure, which is useful for assessing transmission at a population level, as well as historically [46,51,101]. As such, serosurveillance is particularly suited to highlighting areas that may require further intervention and control activities to prevent increased transmission. As seroprevalence results are such a strong indicator of transmission, they could be used to inform malaria transmission and elimination programs, ultimately ensuring resources are targeted to where they are most required [75,83,100,102]. This can also be geared towards preventing future transmission, as malaria patients with long incubation periods could be identified as harbouring parasites and treated accordingly before they are able to contribute to the next peak transmission cycle [72]. For example, in a series of cross-sectional surveys performed by Surendra et al. [76], a region of Indonesia was highlighted by serological testing as an area of concern—this directly preceded an outbreak of malaria within the area. The authors argue that the outbreak may have been avoided if additional interventions had been applied after the serological testing marked it as an area of concern. Furthermore, serosurveillance can be utilised to evaluate the effect of malaria control interventions and inform changes as necessary [82,88]. 

Antibody responses can reliably provide a wealth of information regarding local transmission [47,78,102,103]; however, it is important to note that different environments will generate differing serological patterns. Therefore, factors that may affect the antibody response (such as transmission intensity and population immunological background) must be considered when establishing serosurveillance systems [44,74,101]. It is imperative that optimal serological markers are defined and validated for use in relevant environmental conditions, as the heterogeneity of antibody response may lead some antigens to be advantageous in certain contexts but disadvantageous in others [6,8,44]. Markers with shorter decay periods are more appropriate for measuring recent exposure, whereas those with longer half-lives are more appropriate for assessing any past exposure [39]. For example, utilising longer-lasting antibodies such as *Pv*MSP1-19 and *Pv*AMA1 may be more appropriate in low-endemicity than high-endemicity settings, as they can indicate past exposure has occurred but may not be able to pinpoint when the exposure occurred [44,65]. Additionally, utilising serological markers that are generated in response to vaccination should be avoided in areas where vaccination programs occur to prevent misrepresentation of local transmission [69]. Care must also be taken to ensure results are considered with respect to the local context, including transmission history, immunological background and parasite diversity [101,102,104]. This includes the assumption that, where evidence of past exposure is presumed to indicate historical transmission, there is low resident mobility, and the exposure did not occur external to the local environment [105]. In low transmission contexts, seroprevalence shows greater sensitivity and is less susceptible to seasonal fluctuation than parasite prevalence [65]. Seroconversion rates are a particularly useful measure as they can distinguish between areas of active transmission and areas at low risk of malaria, as well as reconstruct historical transmission patterns [65,75,85]; this is, however, based on the assumption that individuals within a community share historical transmission contexts [75]. Because of this assumption, the measure of seroprevalence in children is particularly effective as they have reduced lifetime exposure compared to adults, are less likely to have experienced varied transmission contexts, and can be used to illustrate the effect of recent interventions on local malaria burden [6,85,86,92]. 

A major obstacle preventing the implementation of serosurveillance for *P. vivax* is the lack of standardisation and validation within the field. There is a need to identify ideal serological markers [8]; however, what is ideal may vary between transmission contexts, and as such, the markers selected for a particular surveillance system must be appropriate with respect to the specific aim of the system. Therefore, standardising various markers across various contexts is an essential step, both for the implementation of serosurveillance systems and the comparability of results across time and regions [78]. Additionally, the strength of an antibody response is measured along a continuum; however, the determination of cut-off points along this continuum to define seropositivity and seronegativity is another key step that can directly shape results [15,49,63,73]. The methodology for this process varies widely, further highlighting the need for standardisation [15,49,73]. The cut-off points can be determined based on true negative and positive sera from parasitologically confirmed cases. The difficulty here lies in accessing these sera from differing transmission contexts and regions, which may not always be available [86]. Finally, after a particular set of serological markers has been chosen for a particular transmission context, the sensitivity and specificity of this set of markers must also be confirmed to ensure the validity of the results [86]. By standardising and validating all components of the serological surveillance system, this process can be optimised and implemented to facilitate malaria elimination [103]. 

The inability to identify and treat asymptomatic individuals is a fundamental gap in the malaria elimination toolkit, but this gap may be filled with serology. One validated serological tool has been proposed by Longley et al. [41]. In this study, the authors explored a panel of 342 proteins to identify markers that were able to indicate recent exposure to *P. vivax*. By measuring antibody responses to these proteins in longitudinal cohort studies, they classified 8 serological markers that represented ideal markers of exposure. Specifically, these chosen markers indicated that a seropositive individual likely had a blood-stage *P. vivax* infection within the last 9 months in low-transmission settings. As most *P. vivax* relapses occur within this time period, these seropositive individuals would likely be carrying the hidden liver-stage hypnozoites [41]. Perhaps more importantly, this panel of markers was then validated in three transmission contexts, and it was determined to have a sensitivity and specificity of 80%. Furthermore, this panel of markers was integrated into a serological test and treatment mathematical model that predicted a 59–69% decrease in *P. vivax* prevalence [41]. This highlights the applicability of this serological marker panel to achieve real outcomes [99]. The serological test and treat approach was compared with mass drug administration and typical screen-and-treat approaches and was found to target a higher proportion of hypnozoite carriers while simultaneously reducing the over-treatment of non-carriers [41]. Although this panel is highly promising, the authors highlighted further possibilities for improvement. This included further purification of protein constructs, exploring various protein expression systems, and assessing antigenic diversity and strain specificity, and indicated further refinement of the process might lead to a smaller panel of serological markers that could be developed into a point-of-care test. Nevertheless, this work provides an example of a validated, built-for-purpose tool specific to the local epidemiological context. Since performing the initial literature search for this scoping review, further work exploring the cross-reactivity between *Plasmodium* species has been published [106]. The authors explored IgG cross-reactivity between *P. vivax* and *P. knowlesi* and found that although *P. knowlesi* induces cross-reactive antibodies to *P. vivax*, these are short-lived, and cross-reactivity can be reduced through appropriate antigen selection. Ultimately a modified panel of 8 serological markers was identified that minimised antibody cross-reactivity, providing another example of a built-for-purpose specific tool. Additionally, this implies that cross-reactivity between other *Plasmodium* species may be minimised or avoided through careful antigen selection. 

Although the scope of this paper is on the scientific barriers and merits of serology implementation, the non-scientific steps and barriers to implementation should be considered as well. This can be difficult, particularly in the context of malaria, as health policies vary globally and are not always well-mapped [107]. Ruwanpura et al. [107] explore the variability in *P. vivax* health policy across seven endemic countries, identify bottlenecks, and make recommendations for the improvement of policy that ultimately will improve malarial elimination efforts. Specifically, highlighted factors include the varied weight given by policymakers to local evidence relevant to malarial programs, the varied weight given to the World Health Organization’s endorsement of malarial programs, and the length of time for policy change to occur, which may be several years. This is clearly a colossal obstacle to the World Health Organization’s goal of malaria elimination by 2030, as the end date is fast approaching. However, rapid policy implementation is possible and has occurred previously, particularly during the COVID-19 pandemic [107]. Although rapid policy change is not faultless nor widespread across disease contexts, it presents a possible step forward for research implementation to have a real-world impact. Additionally, the maintenance of surveillance systems over time is essential in preventing the resurgence or reintroduction of malaria, particularly in pre-elimination regions, as malarial transmission concentrates in high-risk populations that may be more difficult to target [83,103]. As areas move closer to elimination, there may be declining motivation and support for surveillance, which will impede elimination efforts [83,108]. Utilising less labour-intensive methods such as serological testing may assist with this declining motivation, as fewer resources are required to maintain the surveillance system while maintaining the integrity and usefulness of the results. 

It is essential to highlight the strengths and limitations of any conducted research for transparency and to facilitate further improvement. Therefore, the strengths and limitations of this study are indicated here. This is the first study to systematically identify and review research in the field of serological surveillance for *P. vivax* globally. Furthermore, the study adhered to PRISMA-ScR reporting guidelines [27]. As such, the analysis was expected to identify gaps and strengths from previous research, highlighting opportunities for further work in this field. The study’s limitations consist of a lack of critical appraisal of included research, a lack of language inclusivity (as only studies published in English were included), and limited inter-review screening. Similar future studies would benefit from an independent duplicate reviewer screening process to ensure all relevant studies were included, as well as critical appraisal where possible. Language inclusivity is a more complex issue to circumnavigate, as resources involved with translation activities may be costly [109]. However, recent research indicates most research in the biomedical area is published in English [110] and that the exclusion of non-English language research may not significantly impact review results [109]. Furthermore, promising research has been conducted evaluating the ability of English-speaking reviewers to assess non-English language studies for adherence to eligibility criteria, which may eventuate as a means for overcoming the language inclusivity hurdle in review research [111]. Finally, limited inter-reviewer screening occurred during the literature search stage. This may have impacted the list of included studies and the seroprevalence estimates but is unlikely to have overall impacted the identified themes. 

In summary, serosurveillance is an incredibly promising tool that requires further refinement and standardisation. Various combinations of serological exposure markers should be explored to identify panels that are suited for various transmission contexts [38,85]; these panels should then be validated with respect to both the relevant transmission context and surveillance system [44,83,93,94,112]. This process has begun in some instances [41] and should be extended to other serological marker panels. Additionally, serological methods, including the identification of antibody cut-off thresholds, should be standardised where possible [86]. The cross-reactivity of markers and antibodies between species should be further explored and characterised [6,65,70,78,80,87,88]. Furthermore, when broad population health surveys are conducted with blood sample collection, consent forms should include permissive language wherever possible to allow extensive validation of markers across epidemiological contexts [86]. Finally, the belatedness of health policy changes stands to be a prominent obstacle in the face of widespread implementation, and as such, key stakeholders need to collaborate to facilitate change and strive to achieve the World Health Organisation’s malaria goals by 2030.

**Table 1 pathogens-12-00791-t001:** Seroprevalence test method and results for 73 studies. The country of study is also given. Seroprevalence results are given as a percentage of the surveyed population. The studies are ordered by year. Nd = not done. Appendix A summarises the study designs for each paper. Note that antigen names have been taken directly from the published manuscripts for the table below, except for formatting changes, e.g., MSP-1_19_ has been updated to MSP1-19 where necessary.

Study ID	Country of Study	Method	Antigen	Seroprevalence Results
Park 2000 [72]	Korea	ELISA	MSP1-19	15%
Abu-Zeid 2002 [73]	United Arab Emirates	ELISA	MAP4	3.30%
Volney 2002 [49]	Guinea	ELISA; IFA	ELISA: CSP; IFA: blood stages	ELISA: 0–57.4%; IFA: nd
Kim 2003 [90]	Korea	ELISA	CSP1, MSP1, AMA1, SERA, EXP1	7.2% reacted to at least one antigen
Lee 2003 [101]	Korea	ELISA	CSP	0.9–9.6% across regions
Chang 2004 [70]	East Timor	ELISA	CSP, MSP	CSP: 5.7%; MSP: 3.3%
Lim 2005 [48]	Korea	ELISA	CSP	0–10% across regions
Curado 2006 [52]	Brazil	ELISA (IgG); IFA (IgM and IgG)	ELISA: CSP; IFA: blood stages	ELISA: 8.38–34.9% across areas; IFA (IgG): 32.0, 49.0%%, (IgM): 0, 1.93%
Arruda 2007 [54]	Brazil	ELISA	CSP	Up to approx. 70%
Cerutti Jr 2007 [95]	Brazil	ELISA; IFA	ELISA: CSP; IFA: blood stages	ELISA: 25.4% VK210, 6.3% VK247, 10.7% Pv-like; IFA: 6.2% IgM, 37.7% IgG
Ladeia-Andrade 2007 [53]	Brazil	ELISA	MSP1-19	Dry season: 64.0%; Wet season 69.6%
Suárez-Mutis 2007 [113]	Brazil	ELISA	MSP1-19	46.90%
Gomes 2008 [61]	Brazil	ELISA; IFA	ELISA: CSP; IFA: blood stages	ELISA: up to 38% IFA: 45%
Culleton 2009 [87]	Republic of the Congo	ELISA	CSP, MSP1	CSP: 6%; MSP: 10%
Bousema 2010 [65]	Somalia	ELISA	MSP1, AMA1	19.3% reacted to at least one antigen
Cook 2010 [91]	Vanuatu	ELISA	MSP1, AMA1	MSP1: 6.2, 12.6% AMA1: 10.1, 15.0%
Lee 2011 [114]	Korea	IFAT	Whole blood antigen	2.16%
Yildiz Zeyrek 2011 [56]	Turkey	ELISA	MSP1, AMA1-ecto, SERA4, CSP	79.1% responded to at least one antigen
Cook 2012 [81]	Cambodia	ELISA	MSP1-19	August: 7.9%; November 6.0%
Khaireh 2012 [115]	Djibouti	Multiplex bead assay	MSP1	17.50%
Kim 2012 [116]	Korea	IFAT	Whole blood antigen	7.24%
Zoghi 2012 [55]	Iran	ELISA	MSP1-19	0.45–1% across regions and surveys
Cho 2013 [66]	Korea	ELISA	CSP	3.08–50% across regions and years (2010–2011)
Rosas-Aguirre 2013 [78]	Peru	ELISA	MSP1-19	13.60%
Cunha 2014 [117]	Brazil	ELISA	MSP1, AMA1	52.5% to at least one antigen
Fru-Cho 2014 [71]	Cameroon	Rapid immunochromatographic card assay	CSP, MSP	1.1% to at least one antigen
Hristov 2014 [64]	Brazil	ELISA, Immunochromatographic test (ICT)	ELISA: MSP1-19; ICT: CSP, MSP	ELISA: 44%; ICT: 38.4%
Kim 2014a [68]	Korea	IFAT	blood-stage parasites	0.94%
Kim 2014b [77]	Korea	ELISA	MSP1	8.08%
Nam 2014 [57]	Korea	Rapid diagnostic test	CSP, MSP1	CSP: 57.0%; MSP1: 80.2%
Ashton 2015 [112]	Ethiopia	ELISA	MSP1, AMA	11.1% to at least one antigen
Lee 2015 [102]	Korea	ELISA	CSP	6.37%
Piperaki 2015 [50]	Greece	ELISA	CSP, MSP1	0% local residents; 11.8% immigrants
Rosas-Aguirre 2015 [75]	Brazil	ELISA	MSP1, AMA1	33.60%
Lopez-Perez 2016 [62]	Colombia	ELISA; IFA	ELISA: CSP, MSP1 IFA: blood stage antigens	ELISA: CSP (32.4%), MSP1, (55.9%); IFA: 47.1%
Poirier 2016 [97]	Benin	ELISA	CSP1, MSP1	MSP1: 28.7%, CSP1: 21.6%, both: 15.2%
Priest 2016 [86]	Cambodia	Multiplex bead assay	MSP1-19	4.60%
Spring 2016 [89]	Cambodia	ELISA	MSP1	73%
Wahid 2016 [118]	Pakistan	ELISA	MSP1, AMA1	17.6–47.5% across camps
Wangroongsarb 2016 [119]	Thailand	ELISA	MSP1-19, MSP2, CSP, AMA	Urban: 3%; Rural: 15% (to at least one antigen)
Zakeri 2016 [79]	Iran	ELISA	MSP1, AMA1	City: 7%; Village: 13%; (to at least one antigen)
Dewasurendra 2017 [80]	Sri Lanka	ELISA	MSP1, AMA1	63.8, 65.1% across regions to at least one antigen
Ghinai 2017 [120]	Myanmar	ELISA	MSP1, AMA1	3.10%
Idris 2017 [88]	Vanuatu	ELISA	Crude schizont extract, MSP1-19, AMA1	up to 40% across antigens and years
Niang 2017 [92]	Senegal	ELISA	MSP1	58%
Rogier 2017 [67]	Mali	Multiplex bead assay	MSP1-19	17.40%
Sáenz 2017 [7]	Ecuador	ELISA; IFAT	ELISA: CSP, MSP1, IFAT: blood stage antigens	ELISA: CSP, 23.08%, MSP1, 27.23%; IFAT: individual *Pv* results n/a
Seol 2017 [121]	Korea	ELISA	GDH	10.39%
Tadesse 2017 [94]	Ethiopia	ELISA	MSP1, AMA1	8.5–36.3% across regions and surveys
Yalew 2017 [82]	Ethiopia	ELISA	MSP1, AMA1	21.8% (age-adjusted)
Kattenberg 2018 [63]	Vietnam	ELISA	MSP1, AMA1	Mixed models: 24.9% in the final survey; Classification and regression tree method (CART): 34.9% in the final survey
Nyunt 2018 [69]	Myanmar	Protein microarray	MSP1-19, AMA1, DBPII	MSP1-19: 31.5%, AMA1: 24.1%, DBPII: 59.3%
Pereira 2018 [74]	Brazil	ELISA	CSP	62%
Assefa 2019 [85]	Ethiopia	Multiplex bead assay	MSP-1, AMA1	25.00%
Feleke 2019 [59]	Ethiopia	Multiplex bead assay	MSP1-19	50%
Keffale 2019 [84]	Ethiopia	ELISA	AMA1	13.00%
Miguel 2019 [122]	Brazil	ELISA	MSP1-19	7.70%
Surendra 2019 [103]	Indonesia	ELISA	MSP-1, AMA1	1.97% to at least one antigen
Costa 2020 [51]	Brazil	ELISA	MSP1-19	2010: 93.4%, 2012: 78.3%, 2013: 85.1%
Labadie-Bracho 2020 [123]	Suriname	Multiplex bead assay	MSP1-19	Up to approx. 12% across regions
Lee 2020 [93]	Korea	Protein array	LSA-N, CSP-VK210, MSP1-19	6.7–23.8% (by region and antigen)
Lu 2020 [124]	Bangladesh	Multiplex bead assay	MSP1	3.10%
Seck 2020 [58]	Senegal	Multiplex bead assay	MSP1-19	0.7% in the total study population; by age group ranged from 0–1.7%
Surendra 2020 [76]	Indonesia	Multiplex bead assay	AMA1, MSP1-19, EB, RBP1a, RBP2b	38.8–46.3% across surveys
Chotirat 2021 [9]	Thailand	Multiplex bead assay	23 proteins	2.5–16.8% across proteins
Edwards 2021 [105]	Myanmar	ELISA	MSP-1, AMA1	3–19.5%
Lee 2021 [125]	Korea	ELISA	CSP	2017: 1.62%, 2018: 1.22%
Monteiro 2021 [60]	Brazil	Multiplex bead assay	MSP1	52.58%
O’Flaherty 2021 [83]	Myanmar	ELISA	AMA1	28.40%
Leonard 2022 [100]	Ethiopia	Multiplex bead assay	AMA1, MSP1, chimeric MSP1	39.90%
Oviedo 2022 [30]	Haiti	Multiplex bead assay	MSP1-19	0.46%
San 2022 [126]	Vietnam	ELISA	AMA1, MSP1-19, CSP allelic variant 210, CSP allelic variant 247	31.10%
Yao 2022 [15]	China-Myanmar border	ELISA	MSP1-19	6.12–12.41% by region

## Figures and Tables

**Figure 1 pathogens-12-00791-f001:**
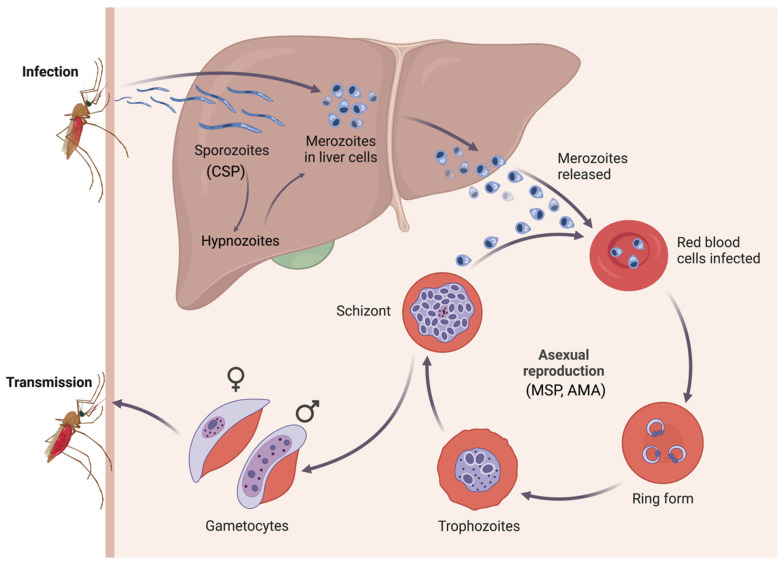
Life cycle diagram illustrating the various stages and transmission cycle of *P. vivax,* including the arrested hypnozoite stage. An infected female *Anopheles* mosquito injects sporozoites into the bloodstream, which migrate to the liver. Sporozoites then develop into hypnozoites (latent infection) and/or merozoites (active infection). Hypnozoites remain dormant in the liver until they reactivate as merozoites weeks to months later. Merozoites released from the liver infect new red blood cells and undergo asexual reproduction to produce the ring stage, trophozoites, and the schizont, as well as gametocytes. The gametocytes are then taken up by another mosquito, and the cycle continues. The most commonly-used antigens in serosurveillance are included in the diagram to indicate where they are expressed: circumsporozoite stage protein (CSP) (sporozoite stage), merozoite surface protein (MSP) (blood stage) and apical membrane antigen (AMA) (blood stage). Created with BioRender.com (Accessed on 19 May 2023) [13].

**Figure 2 pathogens-12-00791-f002:**
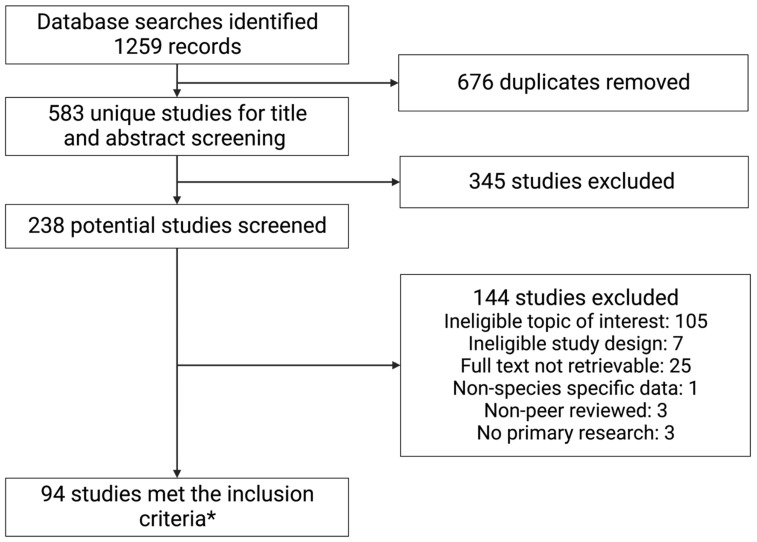
PRISMA flow diagram of study selection. * For details of included studies, see supplementary information. Created with BioRender.com (Accessed on 19 May 2023) [13].

## Data Availability

No new data were created or analyzed in this study. All data used is presented in the manuscript. Data sharing is not applicable to this article.

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
