# Peer review of "Using Serological Markers for the Surveillance of Plasmodium vivax Malaria: A Scoping Review"

_pathogens, 2023, doi:10.3390/pathogens12060791_

Round 1

Reviewer 1 Report

This is a nice review, I only had a few minor comments:

Minor

Line 25 do the authors mean short vs long-term transmission changes or exposure changes. Could be transmission but not sure.

Line 72 clarify what is meant by “this ability is  under utilized”

Line 284-287. This is a very loaded sentence, are the implications that one should expect the prevalence to decrease over time but because the sensitivity is improving that may be confounding that expected trend? This analysis seems a little simplistic. It may be possible but the reason this is not observed could be due to any number of other confounders in the data sets used, including different types of assays, markers, platforms and approaches to differentiating negative and positives. Suggest just stating the observation but not trying to explain it without further analysis.

Lines 303-305: The sentence starting: “As seroprevalence results are…” maybe missing the word “use” between “..adapted to” and “..these results..”. Or alternately it needs some editing.

Line 391: “Ultimately a panel of 8 serological markers…” is this panel distinct from the panel of eight markers describe in line 368? Please clarify.

Line  450: “the lethargy of health”… lethargy is not an appropriate word, most ministries of health as well as the WHO are overcommitted and under-resourced, eliminating P.vivax is a fairly low priority in context of the broader malaria disease burden and for MOHs. The definition of lethargy is” a lack of energy and enthusiasm”. I recommend changing the word while calling out the protracted timelines for policy change.

Reviewer 2 Report

Using serological markers for the surveillance of Plasmodium vivax malaria: a scoping review by Kartal et al.

 This systematic review describes the advantages and disadvantages of serosurveillance tests and markers for P. vivax in various settings, which is an important contribution that summarizes and contrasts serological tests and antigens being used. It is very timely since different regions seek to advance toward eliminating malaria.  Therefore, it is highly relevant to push the development and standardization of sensitive serosurveillance tests and outstanding their potential. The objective is clear and the manuscript is well written, covering the analysis of studies of more than 20 years, since the year 2000. In my opinion, it is suitable for publication in pathogens, only a few minor suggestions are to review.

From the abstract: It is not clear what it means “Seroprevalence results were also recorded where available.”

Line 38: if it is a reference “(World Health Organization, 2021).” please change indicating the number

Please uniform the terminology of the protein markers in Table 1 and the main text.

Table 1:  MSP1-19 or MSP119 , AMA1 or AMA-1, What is CSP-1? , MSP refers to MSP1?,  AMA means AMA1, etc.

Main Text, lines 234 and 325: “ PvMSP-119” please be consistent

Line 147: revise MSP1-1919

 Line 15, 143, and 144. Numbers at the beginning of a sentence should be in letters: Ninety-four (94), Nine (9), or Fifty-three (53)
